# Associations among Perceived Sexual Stigma from Family and Peers, Internalized Homonegativity, Loneliness, Depression, and Anxiety among Gay and Bisexual Men in Taiwan

**DOI:** 10.3390/ijerph19106225

**Published:** 2022-05-20

**Authors:** Huang-Chi Lin, Chih-Cheng Chang, Yu-Ping Chang, Yi-Lung Chen, Cheng-Fang Yen

**Affiliations:** 1Department of Psychiatry, School of Medicine, College of Medicine, Kaohsiung Medical University, Kaohsiung 80708, Taiwan; cochigi@kmu.edu.tw; 2Department of Psychiatry, Kaohsiung Medical University Hospital, Kaohsiung 80756, Taiwan; 3Department of Psychiatry, Chi Mei Medical Center, Tainan 70246, Taiwan; rabiata@mail.chimei.org.tw; 4Department of Health Psychology, College of Health Sciences, Chang Jung Christian University, Tainan 71101, Taiwan; 5School of Nursing, The State University of New York, University at Buffalo, New York, NY 14214-3079, USA; yc73@buffalo.edu; 6Department of Healthcare Administration, College of Medical and Health Science, Asia University, Taichung 41354, Taiwan; 7Department of Psychology, College of Medical and Health Science, Asia University, Taichung 41354, Taiwan; 8College of Professional Studies, National Pingtung University of Science and Technology, Pingtung 91201, Taiwan

**Keywords:** psychological well-being, sexual minority, stigma, internalized homonegativity, loneliness

## Abstract

This study aimed to examine the moderating factors of the association between perceived sexual stigma from family and peers and internalized homonegativity, as well as to compare the effects of perceived sexual stigma from family and peers and internalized homonegativity on loneliness, depression, and anxiety in gay and bisexual men. In total, 400 gay and bisexual men participated in this study. The experiences of perceived sexual stigma from family and peers on the Homosexuality subscale of the HIV and Homosexuality Related Stigma Scale, internalized homonegativity on the Measure of Internalized Sexual Stigma for Lesbians and Gay Men, loneliness on the UCLA Loneliness Scale (Version 3), depression on the Center for Epidemiological Studies-Depression Scale, and anxiety on the State subscale of the State-Trait Anxiety Inventory were collected. The results indicated that perceived sexual stigma from family and peers was significantly associated with internalized homonegativity in both gay and bisexual men, and that sexual orientation moderated the association. Moreover, the association between internalized homonegativity and loneliness was significantly greater than that between perceived sexual stigma from family and peers and loneliness, although no significant differences were observed in their associations with depression and anxiety. Intervention programs that promote changes in the attitudes toward gay and bisexual men among the general population are needed to help prevent the development of internalized homonegativity and further loneliness, depression and anxiety.

## 1. Introduction

Lesbian, gay, and bisexual (LGB) individuals experience multiple forms of sexual stigma due to their sexual orientations (e.g., prejudice; discrimination; and physical, verbal, and relationship bullying rooted in heterosexualism and institutional stigma at legal, social, and cultural levels) [1,2,3]. According to socio-ecological theory [4], LGB individuals may perceive sexual stigma from their microsystems (e.g., family, peers, schools, and health services), exosystems (e.g., neighborhoods, workplaces, and mass media), and macrosystems (e.g., cultures). From a social psychological perspective [5], LGB individuals may endorse perceived sexual stigma from others and develop internalized homonegativity [6]. Because perceived sexual stigma and internalized homonegativity are considerable stressors for LGB individuals [6], they require in-depth investigation.

Family and peers comprise the social microsystems in which LGB individuals are embedded, and they can profoundly influence LGB individuals’ health [7,8,9,10,11]. Sexual stigma from family and peers indicates the ignorance, prejudice and discrimination enacted by family members and peers toward sexual minorities [12,13]. Sexual stigma from family and peers may manifest through a variety of negative attitudes and behaviors, including keeping silent about sexual orientation, sexual orientation-related rejection, bullying, and harassment [12,13]. Sexual stigma from family and peers may compromise health outcomes in LGB individuals. For example, a study in the United States of America (USA) found that young gay and bisexual men reported that family rejection due to sexual orientation decreased instrumental and emotional support and increased the risk of participating in risky ways to search for support, such as engaging in survival sex [12]. Another study in the USA found that family rejection due to sexual orientation during adolescence increased the risks of attempted suicide, depression, illegal drugs use, and engagement of unprotected sexual intercourse in young adult gay and bisexual men [13]. Several studies in the USA have also found that peer bullying due to sexual orientation during adolescence also predicted risky health behaviors, and poor mental and physical health that may last into adulthood [14,15,16,17,18,19,20]. Because family members and peers are the people that LGB individuals most often come into contact with in their daily lives, LGB individuals are extremely likely to internalize sexual stigma they perceive from these people [21,22]. Moreover, family members’ and peers’ negative reactions to LGB individuals’ confirmation of their sexual orientation can exacerbate these individuals’ internalized homonegativity [21,23].

Although perceived sexual stigma has been reported to be the origin of internalized homonegativity in LGB individuals [3,24,25,26], internalized homonegativity may not completely correlate with such individuals’ perceived sexual stigma. For example, a study in the USA on young men who have sex with men (MSM) revealed that the correlation coefficient (r) of the cross-sectional correlation between perceived sexual stigma and three components of internalized homonegativity ranged from 0.07 to 0.19 [27]. This indicates that factors may moderate the association between perceived sexual stigma and internalized homonegativity in LGB individuals. A follow-up study in the USA identified several factors, such as a bisexual sexual orientation, low femininity and high masculinity, severe homophobic bullying victimization, and low peer support, that could predict the non-remission of internalized homonegativity among young adult MSM over a 2-year follow-up period [28]. A cross-sectional study in the USA also revealed stronger associations between homophobic bullying victimization and severity of alcohol use in bisexual boys compared with gay boys [29]. Research has indicated that even when it originates from perceived sexual stigma, the level of internalized homonegativity and its relationship with adverse health outcomes may be moderated by other factors. However, no study has yet examined the moderating effects of age, education level, and sexual orientation on the association between perceived sexual stigma from family members and peers and internalized homonegativity in gay and bisexual men. Identifying the moderators may enable an understanding of the development of internalized homonegativity among LGB individuals.

As two common minority stressors [6], both perceived sexual stigma and internalized homonegativity may negatively affect mental and physical health and relationship functioning; however, mixed results have been provided for these factors. A cross-sectional study in Chile revealed that although both internalized homonegativity and perceived sexual stigma were associated with anxiety and depressive symptoms among sexual minority individuals, only internalized homonegativity was associated with life satisfaction [30]. Another cross-sectional study in the USA demonstrated that internalized homonegativity was associated with poor mental health, whereas perceived sexual stigma was associated with a higher prevalence of sexually transmitted infections, but not with mental health [27]. A prospective study on gay men in the USA revealed that changes in internalized homophobia significantly predicted HIV risk behaviors, and expectations of rejection predicted depressive symptoms [31]. A meta-analysis demonstrated that the effect size of internalized homophobia on same-sex relationship well-being was significantly larger than that of heterosexist discrimination [32]. Another meta-analysis demonstrated that internalized homonegativity was more significantly associated with relationship dysfunction than perceived sexual stigma was [24].

In addition to the differences in the influence of perceived sexual stigma from family and peers and of internalized homonegativity on depression and anxiety, whether perceived sexual stigma from family and peers and internalized homonegativity influence loneliness among gay and bisexual men warrants study. Loneliness is a subjective feeling of perceived discordance between the desired and actual degree of social connectivity [33]. Loneliness is prevalent among gay and bisexual men [34,35,36]. Loneliness can increase the risks of poor physical health [37], sexual risk behaviors [38], and mental health problems [39] in gay and bisexual men. Research has revealed that internalized homonegativity and experiences of sexual orientation discrimination can lead to a higher likelihood of loneliness in gay and bisexual men [40]. However, the different influences of perceived sexual stigma from family and peers and internalized homonegativity on mental health in gay and bisexual men have not been examined.

Research found that tolerance to homosexuality in Taiwan has outpaced that which is found in China, Japan, and South Korea over the past two decades [41]. Liberal values related to divorce, prostitution, and gender roles have been considered as mediators for cohort improvement in tolerant attitudes toward homosexuality in Taiwan [42]. However, sexual orientation bullying [43,44] and microaggression and internalized homonegativity [45] are still common in Taiwan. People in Taiwan have shown their discriminant attitudes toward sexual minority individuals during the debate on legalizing same-sex relationships [46,47,48,49,50,51,52]. Living in such an unfriendly environment, a high proportion of gay and bisexual men in Taiwan suffer from compromised quality of life [53], depression [54], anxiety [55], suicide [55], alcohol [56] and illicit drug use [57]. Therefore, it is important to evaluate the experiences of sexual stigma and mental health problems among gay and bisexual men in Taiwan.

This study on stigma and mental health among gay and bisexual men in Taiwan had two aims. First, we examined the moderating effects of age, educational level, and sexual orientation on the association between perceived sexual stigma from family and peers and internalized homonegativity in gay and bisexual men. Second, we compared the effects of perceived sexual stigma from family and peers and internalized homonegativity on loneliness, depression, and anxiety. We hypothesized that age, educational level, and sexual orientation would moderate the association between perceived sexual stigma from family and peers and internalized homonegativity, and that internalized homonegativity would have a greater effect on loneliness, depression, and anxiety in gay and bisexual men than perceived sexual stigma from family and peers would.

## 2. Materials and Methods

### 2.1. Participants

In the present study, participants were recruited by posting advertisements from August 2021 to January 2022 on the Bulletin Board System (a popular application for online message sharing in Taiwan), Facebook, LINE (a popular messaging app), and the web sites of three health promotion centers for sexual minority individuals. Taiwanese gay or bisexual men who resided in Taiwan were included in this study. The exclusion criterion was difficulty in comprehending the purpose or questionnaire content of this study due to intellectual disability and cognitive dysfunction caused by alcohol and substance use or brain injury. Regarding the sample size, according to VanVoorhis and Morgan [58], at least 30 participants per variable are needed to detect a small effect size in the regression equations. There were five independent variables (age, education level, sexual orientation, perceived sexual stigma, and internalized homonegativity) and six interaction variables (interactions between demographic characteristics and sexual stigma) in this study; therefore, there needed to be at least 330 participants to detect a small effect size in the regression equations. In total, 400 gay and bisexual male participants were included and provided written informed consent. The participants completed the paper-and-pencil questionnaires in the study rooms of the psychiatry research department affiliated to a university hospital in person and were assured that their responses to the research questionnaire would be confidential. The participants were allowed to seek help from research assistants when they experienced difficulty in completing the questionnaire. The Institutional Review Board of Kaohsiung Medical University Hospital approved the study (KMUHIRB-F(I)-20210119).

### 2.2. Measures

#### 2.2.1. Perceived Sexual Stigma from Family and Peers

We used the 10-item Homosexuality subscale of the HIV and Homosexuality Related Stigma Scale (HHRS) [59] to assess perceived stigmatizing attitudes toward homosexuality from families and peers (e.g., “My family and peers have negative attitudes toward homosexuality” and “My family and peers would be disappointed to have a gay son or friend”). The HHRS was developed and validated by a study in China to assess HIV and homosexuality-related stigma among gay and bisexual men but not limited to the individuals with HIV [59]. The participants rated each item on a 4-point Likert type scale, with scores of 1, 2, 3, and 4 indicating “strongly disagree”, “disagree”, “agree”, and “strongly agree”, respectively. Those with a higher total score had a higher level of perceived sexual stigma. The HHRS-Homosexuality subscale was reported to have satisfactory reliability (Cronbach’s α = 0.85) and psychometric properties [59]. The Cronbach’s α of the HHRS-Homosexuality subscale for this sample was 0.92.

#### 2.2.2. Internalized Homonegativity

We used the Mandarin Chinese version [60] of the 17-item Measure of Internalized Sexual Stigma for Lesbians and Gay Men (MISS-LG) [61] to assess participants’ internalized homonegativity. Two versions of the MISS-LG have been developed, which are as follows: one for lesbian and bisexual women and one for gay and bisexual men; the version for gay and bisexual men was used in this study (e.g., “I would not tell my friends that I am gay because I would be afraid of losing them” and “I would prefer to be heterosexual”). The participants rated each item on a 5-point Likert type scale, with scores of 1, 2, 3, 4, and 5 indicating “strongly disagree”, “disagree”, “neither agree nor disagree”, “agree”, and “strongly agree”, respectively. The participants with a higher total score had a higher level of internalized homonegativity. The psychometric properties of the MISS-LG were reported to be satisfactory [61]. A previous study also supported its psychometric properties and satisfactory internal consistency (Cronbach’s α = 0.90) among sexual minority individuals in Taiwan [60].

#### 2.2.3. Anxiety

We used the Mandarin Chinese version of the 20-item State subscale [62] of the State-Trait Anxiety Inventory (MC-STAI-S) [63] to assess current anxiety symptoms (e.g., “I feel nervous” and “I feel jittery”). The participants rated each item on a 4-point Likert scale, with scores of 1, 2, 3, and 4 indicating “not at all”, “a little”, “somewhat”, and “very much”, respectively. Those with a higher total score had more severe current anxiety. The MC-STAI-S was reported to have acceptable reliability (test–retest reliability: Pearson’s correlation r = 0.76; internal consistency: Cronbach’s α = 0.91), criterion validity (correlation with the Hamilton Anxiety Rating Scale: Pearson’s correlation r = 0.69), and construct validity [64]. The internal consistency of the MC-STAI-S was satisfactory in the present study (Cronbach’s α = 0.95).

#### 2.2.4. Depression

We used the 20-item Mandarin Chinese version [65] of the Center for Epidemiological Studies-Depression Scale (MC-CES-D) to assess the frequency of depressive symptoms in the month preceding the study (e.g., “I felt depressed” and “My sleep was restless”) [66]. The participants rated each item on a 4-point scale, with scores of 1, 2, 3, and 4 indicating “rarely or none of the time”, “some of the time”, “occasionally or a moderate amount of the time”, and “most or all of the time”, respectively. Those with a higher total score had more severe depressive symptoms. The MC-CES-D was reported to have good reliability (internal consistency: Cronbach’s α = 0.90; 1-week test–retest reliability: intraclass correlation reliability = 0.93), congruent validity (with the diagnosis of major depressive disorders) [67], and construct validity [68]. The internal consistency of the MC-CES-D was satisfactory in the present study (Cronbach’s α = 0.92).

#### 2.2.5. Loneliness

We used the 20-item Chinese version [59] of the UCLA Loneliness Scale, Version 3, to assess participants’ current feelings of loneliness (e.g., “I lack companionship” and “My interests and ideas are not shared by those around me”) [45]. The use of the UCLA Loneliness Scale (Version 3) has been widely supported by much evidence showing its good psychometric properties in different aspects. For example, the scale has robust psychometric properties, such as internal consistency, test–retest reliability, concurrent validity, and construct validity in the Chinese version [69] and many other language versions [70,71,72,73,74] across different populations (e.g., healthy participants, older people, adolescents, and mothers). The participants rated each item on a 4-point Likert scale, with scores of 1, 2, 3, and 4 indicating “never”, “rarely”, “sometimes”, and “always”, respectively. Nine items were reverse coded, and a higher total score indicated a higher level of loneliness. The internal consistency and construct validity of the UCLA Loneliness Scale, Version 3, were reported to be satisfactory [75]; for example, the Cronbach’s α was 0.89 to 0.94. The internal consistency of the UCLA Loneliness Scale, Version 3, was satisfactory in the present study (Cronbach’s α = 0.92).

#### 2.2.6. Demographic and Sexual Orientation Factors

The data of participants’ ages, education levels (high school or below vs. college or above), and sexual orientations (homosexual or bisexual) were collected.

### 2.3. Data Analysis

The participants’ demographic and sexual orientation factors were analyzed using descriptive statistics. The absolute skewness and kurtosis values for the scores of perceived sexual stigma from family and peers, internalized homonegativity, depression, anxiety, and loneliness ranged from 0.202 to 0.773 and 0.087 to 0.582, respectively; according to Kim [76], these scores were normally distributed. The associations between perceived sexual stigma from family and peers, demographics, and sexual orientation and internalized homonegativity were examined using multivariate linear regression analysis. The moderating effects of age, education level, and sexual orientation on the association between perceived sexual stigma from family and peers and internalized homonegativity were also examined based on the criteria proposed by Baron and Kenny [77].

The associations between perceived sexual stigma from family and peers and internalized homonegativity and depression, anxiety, and loneliness were also examined and compared using multivariate linear regression analysis, after controlling for the effects of demographics and sexual orientation. To compare the magnitude of the associations of perceived sexual stigma from family and peers and internalized homonegativity with depression, anxiety, and loneliness, we used the standardized scores of perceived sexual stigma from family and peers and those of internalized homonegativity, depression, anxiety, and loneliness to enable these variables to be considered on the same scale. We reported a standardized regression coefficient (β), which enabled the comparison of the regression coefficients between variables. Finally, we used the Wald test to compare the equality of the distances of the standardized regression coefficients for perceived sexual stigma from family and peers and internalized homonegativity, with respect to depression, anxiety, and loneliness [78]. A value of *p* < 0.05 was considered statistically significant. All the analyses were performed using the IBM SPSS 20.0 (IBM, Armonk, NY, USA).

## 3. Results

The study sample characteristics (N = 400) are listed in Table 1. The mean age of the participants was 30.7 years (standard deviation (SD) = 5.9). Most of the participants were well-educated (83.2% had a college or above educational level) and homosexual (83.2%). The mean scores (SD) for perceived sexual stigma from family and peers and internalized homonegativity were 26.9 (6.8) and 40.8 (12.3), respectively. The mean scores for the severity of depression, anxiety, and loneliness were 18.3 (11.1), 39.2 (12.5), and 43.5 (11.1), respectively.

The results of the multivariate linear regression analysis of the association between the factors related to internalized homonegativity in gay and bisexual men are presented in Table 2. The results of Model I indicate that older age and a higher perceived level of sexual stigma from family and peers were significantly associated with higher internalized homonegativity. Bisexual men had higher internalized homonegativity than gay men did. To examine the moderating effects of age and sexual orientation on the association between perceived sexual stigma from family and peers and internalized homonegativity, the interactions between sexual stigma from family and peers and age and sexual orientation were included in Model II. The results indicate that the interaction between perceived sexual stigma from family and peers and sexual orientation was significantly associated with internalized homonegativity. Further analysis revealed that the association between perceived sexual stigma from family and peers and internalized homonegativity was greater in bisexual men (B = 0.271, se = 0.218, *p* < 0.001) than in gay men (B = 0.630, se = 0.086, *p* < 0.001).

The results of the multivariate linear regression analysis of the associations of perceived sexual stigma from family and peers and internalized homonegativity with depression, anxiety, and loneliness are presented in Table 3. The results indicate that after controlling for age, education level, and sexual orientation, both perceived sexual stigma from family and peers and internalized homonegativity were significantly and positively associated with depression, anxiety, and loneliness. We further compared the equality of the distance between the standardized regression coefficients for perceived sexual stigma from family and peers and internalized homonegativity, with respect to depression (0.241 vs. 0.235), anxiety (0.129 vs. 0.243), and loneliness (0.173 vs. 0.339). The standardized regression coefficient for internalized homonegativity was significantly greater than that of perceived sexual stigma from family and peers, with respect to loneliness (*p* = 0.047), although such differences were not observed with respect to depression and anxiety (*p*s > 0.05).

## 4. Discussion

The present study revealed that perceived sexual stigma from family and peers was significantly associated with internalized homonegativity in both gay and bisexual men, and that sexual orientation moderated the association. Moreover, the association between internalized homonegativity and loneliness was significantly greater than that between perceived sexual stigma from family and peers and loneliness, although no significant differences were observed in their associations with depression and anxiety.

The results of this study support the social psychological hypothesis that perceived sexual stigma from people close to gay and bisexual individuals is significantly associated with internalized homonegativity among such individuals [5,6]. According to minority stress theory [6], gay and bisexual men may perceive negative social values toward the self and develop internalized homonegativity, even in the absence of overt negative events. Thoits described such a process of internalized stigma, explaining that “role-taking abilities enable individuals to view themselves from the imagined perspective of others. One can anticipate and respond in advance to others’ reactions regarding a contemplated course of action” [79]. Family and peers comprise the social microsystems in which gay and bisexual men are embedded; gay and bisexual men may have a lot of opportunity to experience and are heavily influenced by sexual stigma from family and peers. The cross-sectional study design limited our ability to determine the temporal relationship between perceived sexual stigma from family and peers and internalized homonegativity. However, gay and bisexual men may feel ashamed and conceal their sexual orientation after being exposed to negative public attitudes toward sexual minorities [5]. Moreover, internalized homonegativity may increase gay and bisexual men’s sensitivity to family members’ and peers’ comments regarding sexual orientation, which can then increase the severity of perceived sexual stigma. The results of the present study support the importance of family education and cultural change aimed at reducing public stigma against sexual minorities to mitigate internalized homonegativity among gay and bisexual men.

Previous studies have revealed that bisexual men have greater internalized homonegativity than gay men do [80,81,82,83,84]. The present study further revealed that the association between perceived sexual stigma from family and peers and internalized homonegativity was greater in bisexual men than in gay men. According to socio-ecological theory [4], internalized homonegativity is the result of interactions among individuals and their microsystems, exosystems, and macrosystems. Bisexual individuals are often accused of sexual orientation instability and sexual irresponsibility from lesbians, gay men, and heterosexual individuals [85,86,87]; therefore, bisexual men may develop internalized homonegativity as a part of their internal value systems and in relation to identity [88]. The results of this study indicate that the moderating role of sexual orientation should be considered in intervention programs to reduce sexual stigma and internalized stigma.

In addition, our results indicated that both perceived sexual stigma from family and peers and internalized homonegativity were significantly associated with depression and anxiety in gay and bisexual men; no significant differences were found in the levels of associations. The results supported minority stress theory that both perceived sexual stigma and internalized homonegativity are minority stressors that may compromise the mental health of gay and bisexual men [6]. Furthermore, according to cognitive theory [89], mood problems may interact with dysfunctional cognitive patterns, such as maladaptive attention and interpretation bias [89]; gay and bisexual men may have a higher awareness of other people’s homonegative attitudes, which further merge with their own internalized homonegativity. Notably, in our study, although both perceived sexual stigma and internalized homonegativity were significantly associated with loneliness, the associations between internalized homonegativity and loneliness were significantly greater than that between perceived sexual stigma from family and peers and loneliness.

According to self-theory [90], the concept of the self is the experiences that individuals label as ‘‘mine’’ as they grow and understand the world; the concept of the self is also considered as the representation of an individual’s self-perceptions, personal attributions, and past experiences. Therefore, the concept of the self could be defined as a set of images one has about himself or herself [90]. Internalized homonegativity may insidiously influence gay and bisexual men’s beliefs regarding interpersonal interactions, even in the absence of others with prejudices; this may cause social disconnection with others across social contexts to develop progressively, subsequently intensifying loneliness. A meta-analysis also found that internalized homonegativity had a more deleterious effect on relationship functioning than perceived sexual stigma did [24]. Moreover, internalized homonegativity may develop over a long period, starting at the initial awareness of sexual orientation. Although adult gay and bisexual men have increased autonomy and can avoid interactions with family and peers who are prejudiced toward sexual minorities, the longstanding internalized homonegativity may continue to influence their satisfaction with the quality and quantity of their social connections with others. Because of the profound influence of loneliness on health [91], intervention programs to reduce sexual stigma in the general population and within family units and to prevent internalized homonegativity are necessary.

There are some limitations in the present study. First, the cross-sectional study design limited the inferences concerning the temporal relationships among perceived sexual stigma from family and peers, internalized homonegativity, loneliness, depression and anxiety. Second, all the data collected in the present study were self-reported. Therefore, single-rater biases and social desirability biases cannot be fully controlled. Third, the present study did not take the possibility of transgender, gender nonbinary, and genderqueer into consideration. Research has indicated that sexual and gender minority identities have intersectional impacts on health [92] and behaviors [93]. Both sexual and gender minority identities should be considered in public health practice [94]. Last, the participants were recruited via online advertisements. Online advertisements can recruit numerous gay and bisexual participants [95,96]; however, gay and bisexual individuals are not equally approached [97]. For example, Facebook users consist of younger people among the general population [97]. Therefore, our sample was not nationally representative.

## 5. Conclusions

The present study revealed that perceived sexual stigma from family and peers was significantly associated with internalized homonegativity in both gay and bisexual men, and that sexual orientation moderated the association. Both perceived sexual stigma from family and peers and internalized homonegativity were significantly associated with loneliness, depression and anxiety, whereas the association between internalized homonegativity and loneliness was significantly greater than that between perceived sexual stigma from family and peers and loneliness. Developing intervention programs to promote changes in the attitudes toward gay and bisexual men among the general population is the necessary step to help prevent the development of internalized homonegativity and further loneliness, depression and anxiety. Meanwhile, given the effect of sexual orientation on the associations between perceived sexual stigma from family and peers and internalized homonegativity identified in this study, intervention programs should take sexual orientation into consideration.

## Figures and Tables

**Table 1 ijerph-19-06225-t001:** Participant characteristics (N = 400).

Variables	Mean (SD)	Range	n (%)
Age (years)	30.7 (5.9)	20–51	
Educational level			
High school or below			67 (16.8)
College or above			333 (83.2)
Sexual orientation			
Homosexual			333 (83.2)
Bisexual			67 (16.8)
Perceived sexual stigma from families and peers	26.9 (6.8)	10–40	
Internalized homonegativity	40.8 (12.3)	18–83	
Depression	18.3 (11.1)	0–58	
Anxiety	39.2 (12.5)	20–72	
Loneliness	43.5 (11.1)	20–80	

**Table 2 ijerph-19-06225-t002:** Factors related to internalized homonegativity: multivariate linear regression analysis.

Variables	Model I			Model II		
	B	se	*p*	B	se	*p*
Age	0.280	0.092	0.003	0.026	0.373	0.944
Education degree of college or above ^a^	0.472	1.475	0.749	0.445	1.464	0.761
Bisexual men ^b^	8.707	1.467	<0.001	−10.273	6.102	0.093
Perceived sexual stigma from families and peers	0.730	0.081	<0.001	0.317	0.421	0.452
Age × Perceived stigmatizing attitudes from families and peers				0.010	0.014	0.453
Sexual orientation × Perceived sexual stigma from families and peers				0.728	0.227	0.001

^a^: education degree of high school or below as reference; ^b^: gay men as reference. se = standard error.

**Table 3 ijerph-19-06225-t003:** Associations between perceived sexual stigma from family and peers and internalized homonegativity and depression, anxiety, and loneliness: multivariate linear regression analysis.

	Depression			Anxiety			Loneliness		
Variable	β	se	*p*	β	se	*p*	β	se	*p*
Perceived sexual stigma from families and peers	0.241	0.051	<0.001	0.129	0.053	0.015	0.173	0.049	<0.001
Internalized homonegativity	0.235	0.053	<0.001	0.243	0.054	<0.001	0.339	0.050	<0.001

β = standardized regression coefficient, se = standard error. Analysis was conducted with adjustment for age, sexual orientation, and educational level.

## Data Availability

The data will be available upon reasonable request to the corresponding authors.

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
