# Peer review of "Associations among Perceived Sexual Stigma from Family and Peers, Internalized Homonegativity, Loneliness, Depression, and Anxiety among Gay and Bisexual Men in Taiwan"

_ijerph, 2022, doi:10.3390/ijerph19106225_

Round 1

Reviewer 1 Report

Thank you for the invitation to review this manuscript. The submitted manuscript tries to elucidate the complex interplay between various variables that influence internalized homophobia and down the line have an effect on the health outcomes of gay and bisexual men. The manuscript is well written and seems to be methodologically sound. There are however a few issues, that I feel may improve the manuscript. 

  1. Introduction: I feel a definition of "perceived sexual stigma from family and peers" is needed. This is a descriptive name and I feel it does not help some of the readers understand what it actually is. Especially as there is a lot of talk of stigmatization and discrimination overall and from other sources in the article. Therefore, I would strongly suggest that all the sociological and psychological phenomena in the manuscript are clearly described. I also feel one should discuss in a bit more detail what health outcomes are associated and how with the variables of interest.
  2. Methods: It is unclear if this is or was a purely online questionnaire? How did you prevent multiple entries? Did you do any analyses post hoc to prevent these? How many times was the questionnaire accessed? How was the response rate? Why did you stop at 400? "Perceived sexual stigma from family and peers" questionnaire that you used: why did you use the subscale of the HHRS as your article was not limited to HIV? Also you do not mention if this was available in Mandarin Chinese, therefore how did you translate it? Similar for the "UCLA Loneliness Scale": I am interested in the translation and in the choice for the scale. 
  3. Results, Discussion and Conclusions seem appropriately written and I have no major comments.

Author Response

We appreciated your valuable comments. As discussed below, we have revised our manuscript with underlines based on your suggestions. Please let us know if we need to provide anything else regarding this revision.

Comment 1

Introduction: I feel a definition of "perceived sexual stigma from family and peers" is needed. This is a descriptive name and I feel it does not help some of the readers understand what it actually is. Especially as there is a lot of talk of stigmatization and discrimination overall and from other sources in the article. Therefore, I would strongly suggest that all the sociological and psychological phenomena in the manuscript are clearly described. I also feel one should discuss in a bit more detail what health outcomes are associated and how with the variables of interest.

Response

Thank you for your comment. We added the definition and health outcomes of perceived sexual stigma from family and peers into the Introduction as below. Please refer to line 55-70.

… Sexual stigma from family and peers indicates the ignorance, prejudice and discrimination enacted by family members and peers toward sexual minorities [12,13]. Sexual stigma from family and peers may manifest through a variety of negative attitudes and behaviors, including keeping silent about sexual orientation, sexual orientation-related rejection, bullying, and harassment [12,13]. Sexual stigma from family and peers may compromise health outcomes in LGB individuals. For example, a study in the United States of America (USA) found that young gay and bisexual men reported that family rejection due to sexual orientation decreased instrumental and emotional support and increased the risk of participating in risky ways to search for support, such as engaging in survival sex [12]. Another study in USA found that family rejection due to sexual orientation during adolescence increased the risks of attempted suicide, depression, illegal drugs use, and engagement of unprotected sexual intercourse in young adult gay and bisexual men [13]. Several studies in USA have also found that peer bullying due to sexual orientation during adolescence also predicted risky health behaviors, and poor mental and physical health that may last into adulthood [14-20].….

Comment 2-1.

Methods: It is unclear if this is or was a purely online questionnaire? How did you prevent multiple entries? Did you do any analyses post hoc to prevent these? How many times was the questionnaire accessed? How was the response rate?

Response

Thank you for your remining. The participants completed the paper-and-pencil questionnaire in person. Therefore, there was no multiple entries in this study. We added the introduction for the method of collecting data form the participants into “2.1. Participants” as below. Please refer to line 161-165.

Participants completed the paper-and-pencil questionnaires in the study rooms of the psychiatry research department affiliated to a university hospital in person and being assured that their responses to the research questionnaire would be confidential. The participants were allowed to seek help from research assistants when they experienced difficulty in completing the questionnaire.

Comment 2-2.

Why did you stop at 400?

Response

Thank you for your comment. We added the method for estimating the sample size needed in this study as below. Please refer to line 154-159.

“Regarding to the sample size, according to VanVoorhis and Morgan [58], at least 30 participants per variable are needed to detect a small effect size in regression equations. There were five independent variables (age, education level, sexual orientation, perceived sexual stigma, and internalized homonegativity) and six interaction variables (interactions between demographic characteristics and sexual stigma) in this study; therefore, it needs at least 330 participants to detect a small effect size in regression equations.”

Comment 2-3.

"Perceived sexual stigma from family and peers" questionnaire that you used: why did you use the subscale of the HHRS as your article was not limited to HIV? Also you do not mention if this was available in Mandarin Chinese, therefore how did you translate it?

Response

Thank you for your comment. The HHRS was developed and validated by a study on Chinese gay and bisexual men to assess HIV and homosexuality related stigma. Therefore, the original version of the HHRS is in Chinese and we do not have to translate it. Meanwhile, the use of the HHRS is not limited to the individuals with HIV. We added the explanation into “2.2.1. Perceived Sexual Stigma from Family and Peers” as below. Please refer to line 173-175.

The HHRS was developed and validated by a study in China to assess HIV and homosexuality related stigma among gay and bisexual men but not limited to the individuals with HIV [59].

Comment 2-4.

Similar for the "UCLA Loneliness Scale": I am interested in the translation and in the choice for the scale. 

Response

Thank you for your comment. We used the well-translated Chinese version of the UCLA Loneliness Scale in this study. We added the introduction for it and the choice for the scale into “2.2.5. Loneliness” as below. Please refer to line 219-226.

We used the 20-item Chinese version [59] of the UCLA Loneliness Scale, Version 3, to assess participants’ current feelings of loneliness (e.g., “I lack companionship” and “My interests and ideas are not shared by those around me”) [45]. The use of UCLA Loneliness Scale (Version 3) has been widely supported by much evidence showing its good psychometric properties in different aspects. For example, the scale has robust psychometric properties such as internal consistency, test-retest reliability, concurrent validity, and construct validity in a Chinese version [69] and many other language versions [70-74] across different populations (e.g., healthy participants, older people, adolescents, and mothers).

Comment 3

Results, Discussion and Conclusions seem appropriately written and I have no major comments.

Response

Thank you for your positive comment.

Reviewer 2 Report

This is a very interesting and timely paper. I compliment the authors for this paper, which concerned with an important topic, filling a gap in the current scientific literature.
Although in my opinion this manuscript should be published, I would suggest some changes for improving it even more, as reported below.
I hope that authors will find my suggestions in a constructive way.

Since the social, cultural and normative climate significantly affects the level of inclusion towards sexual and gender minorities, I suggest that the authors indicate in the introduction where the studies they cite have been conducted.

In the same way, I believe that it is very important to describe the socio-cultural and normative context of research. How do sexual and gender minorities live in Taiwan? What is the political climate towards them? In order to deepen these aspects, it may be useful to cite the statistics and studies of William Institute, ILGA, Spartacus etc.

Finally, I invite the authors to specify why they have chosen to carry out their recruitment exclusively online, also indicating the limits and potential of this technique. In this regard, I suggest them to consult the texts: 

Matthews, J., & Cramer, E. P. (2008). Using Technology to Enhance Qualitative Research with Hidden  Populations. Qualitative Report, 13(2), 301–315

Monaco, S. (2022), ‘Gender and Sexual Minority Research in the Digital Society’, in Punziano G., Delli Paoli A. (Eds.), Handbook of Research on Advanced Research Methodologies for a Digital Society, Hersey, IGI Global, 885-897.

Rait, M. A., Prochaska, J. J., & Rubinstein, M. L. (2015). Recruitment of adolescents for a smoking  study: Use of traditional strategies and social media. Translational Behavioral Medicine, 5(3), 254–259.  doi:10.100713142-015-0312-5 PMID:26327930

Ramo, D. E., & Prochaska, J. J. (2012). Broad reach and targeted recruitment using facebook for an online  survey of young adult substance use. Journal of Medical Internet Research, 14(1), e28. doi:10.2196/jmir.1878 PMID:22360969

Finally, The discussion is full of interesting explanations of the results. However, at the same time, it is quite confusing, as it did not follow hypotheses. I suggest authors to rewrite the discussion on the basis of confirmation/non-confirmation of the hypotheses.

Author Response

We appreciated your valuable comments. As discussed below, we have revised our manuscript with underlines based on your suggestions. Please let us know if we need to provide anything else regarding this revision.

Comment 1

Since the social, cultural and normative climate significantly affects the level of inclusion towards sexual and gender minorities, I suggest that the authors indicate in the introduction where the studies they cite have been conducted.

Response

Thank you for your suggestion. We indicated countries where the studies have been conducted in the introduction. Most of the studies cited in the introduction have been conducted in USA. Please refer to Line 61-104.

Comment 2

In the same way, I believe that it is very important to describe the socio-cultural and normative context of research. How do sexual and gender minorities live in Taiwan? What is the political climate towards them? In order to deepen these aspects, it may be useful to cite the statistics and studies of William Institute, ILGA, Spartacus etc.

Response

We added a new paragraph introducing the socio-cultural background for sexual minority individuals in Taiwan as below. Please refer to line 123-134.

Research found that tolerance to homosexuality in Taiwan has outpaced that which is found in China, Japan, and South Korea over the past two decades [41]. Liberal values related to divorce, prostitution, and gender roles have been considered as mediators for cohort improvement in tolerant attitudes toward homosexuality in Taiwan [42]. However, sexual orientation bullying [43,44] and microaggression and internalized homonegativity [45] are still common in Taiwan. People in Taiwan have shown their discriminant attitudes toward sexual minority individuals during the debate on legalizing the same-sex relationships [46-52]. Living in such an unfriendly environment, a high proportion of gay and bisexual men in Taiwan suffer from compromised quality of life [53], depression [54], anxiety [55], suicide [55], alcohol [56] and illicit drug use [57]. Therefore, it is important to evaluate the experiences of sexual stigma and mental health problems among gay and bisexual men in Taiwan.

Comment 3

Finally, I invite the authors to specify why they have chosen to carry out their recruitment exclusively online, also indicating the limits and potential of this technique.

Response

We added discussion regarding recruiting participants via online advertisements as below. Please refer to line 384-388.

Last, the participants were recruited via online advertisements. Online advertisements can recruit numerous gay and bisexual participants [95,96]; however, gay and bisexual individuals are not equally approached [97]. For example, Facebook users consist of younger people among the general population [97]. Therefore, our sample was not nationally representative.

Comment 4

The discussion is full of interesting explanations of the results. However, at the same time, it is quite confusing, as it did not follow hypotheses. I suggest authors to rewrite the discussion on the basis of confirmation/non-confirmation of the hypotheses.

Response

Thank you for your suggestion. We rewrote the discussion on the basis of the hypotheses as below, including:

According to minority stress theory [6], gay and bisexual men may perceive negative social values toward the self and develop internalized homonegativity even in the absence of overt negative events. Thoits described such a process of internalized stigma, explaining that “role-taking abilities enable individuals to view themselves from the imagined perspective of others. One can anticipate and respond in advance to others’ reactions regarding a contemplated course of action” [79]. Family and peers comprise the social microsystems in which gay and bisexual men are embedded; gay and bisexual men may have a lot of opportunity to experience and are heavily influenced by sexual stigma from family and peers.” Please refer to line 315-323.

According to socio-ecological theory [4], internalized homonegativity is the results of interactions among individuals and their microsystems, exosystems, and macrosystems. Bisexual individuals are often accused of sexual orientation instability and sexual irresponsibility from lesbians, gay men, and heterosexual individuals [85-87]; therefore, bisexual men may develop internalized homonegativity as a part of their internal value systems and in relation to identity [88]. Please refer to line 336-342.

“…according to cognitive theory [89], mood problems may interact with dysfunctional cognitive patterns, such as maladaptive attention and interpretation bias [89]; gay and bisexual men may have a higher awareness of other people’s homonegative attitudes, which further merge with their own internalized homonegativity. Please refer to line 350-354.

According to self-theory [90], the concept of the self is the experiences that individuals label as ‘‘mine’’ as they grow and understand the world; the concept of the self is also considered as the representation of an individual’s self-perceptions, personal attributions, and past experiences. Therefore, the concept of the self could be defined as a set of images one has about himself or herself [90]. Please refer to line 358-362.